# Evolution of Plant Virus Diagnostics Used in Australian Post Entry Quarantine

**DOI:** 10.3390/plants10071430

**Published:** 2021-07-13

**Authors:** Mark Whattam, Adrian Dinsdale, Candace E. Elliott

**Affiliations:** Department of Agriculture, Water and Environment, 135 Donnybrook Rd, Mickleham, VIC 3156, Australia; adrian.dinsdale@awe.gov.au (A.D.); candace.elliott@awe.gov.au (C.E.E.)

**Keywords:** plant quarantine, biosecurity, plant imports, virus diagnostics, post-entry quarantine, phytosanitary

## Abstract

As part of a special edition for MDPI on plant virology in Australia, this review provides a brief high-level overview on the evolution of diagnostic techniques used in Australian government Post-Entry Quarantine (PEQ) facilities for testing imported plants for viruses. A comprehensive range of traditional and modern diagnostic approaches have historically been employed in PEQ facilities using bioassays, serological, and molecular techniques. Whilst these techniques have been effective, they are time consuming, resource intensive and expensive. The review highlights the importance of ensuring the best available science and diagnostic developments are constantly tested, evaluated, and implemented by regulators to ensure primary producers have rapid and safe access to new genetics to remain productive, sustainable and competitive.

## 1. Introduction

The expansion of modern and rapid international transport systems together with ever increasing trade and consumer demand has increased the global movement of plants and plant products from their centres of origin. Whilst this has provided numerous benefits for society in expanding crop production and food availability, it has also provided pathways for long-distance spread of plant pests (including invertebrates, pathogens and weed seed propagules) as it is difficult to trade plants and plant products without creating a potential biosecurity risk for the importing country.

Protecting Australia’s biosecurity is a responsibility shared by government, industry, and the community. To maintain Australia’s favourable pest and disease biosecurity status, a ‘systems approach’ is adopted whereby a series of measures are adopted across the biosecurity continuum including offshore, at border, and onshore with the collective aim of preventing or minimising the introduction and/or spread of harmful organisms. Under the Sanitary and Phytosanitary Measures (SPS) agreement, as a member of the World Trade Organization (WTO), Australia has adopted an ‘appropriate level of protection’ (ALOP) defined as providing a high level of biosecurity protection aimed at reducing risk to a very low level, but not to zero, which is unattainable in practical terms.

Rapid and safe access to new plant genetic stocks is crucial for Australia’s plant primary industries to remain profitable, sustainable, and internationally competitive. In Australia, the Department of Agriculture, Water, and the Environment (DAWE) regulates and sets import conditions for the importation of all live plants and seeds for sowing. Import conditions vary depending on the genus and species of plant, the country of origin and the form of plant material imported (for example, tissue cultured plants imported from approved tissue culture suppliers typically pose a lower biosecurity risk compared to cuttings of the same species collected from the field). Full details of conditions for the importation of live plants and seeds into Australia can be found in the department’s Biosecurity Import Conditions (BICON) database (https://bicon.agriculture.gov.au/BiconWeb4.0) (accessed on 8 July 2021).

Plants imported into Australia must be healthy, free of insect pests and show no signs of disease to meet inspection and clearance requirements. The exporting country’s national plant protection organisation or their authorized agent inspects the plants and issues a phytosanitary certificate indicating the plants are free from harmful pests and diseases prior to export. BICON and import permits may also specify additional declarations on the phytosanitary certificate. On arrival, imported plants are inspected by DAWE biosecurity officers for signs of pests or diseases. Imported plants are then released, directed for treatment and/or sent to post-entry quarantine (PEQ) facilities for pathogen screening and/or testing depending on the level of biosecurity risk posed. DAWE recognises high health-planting sources overseas as centres of excellence (e.g., the Scottish Seed Potato Classification Scheme). These centres provide a high level of compliance with regulatory requirements including scientific integrity in pest and disease screening and other biosecurity processes. The post-entry phytosanitary requirements for certain plant material sourced from these suppliers can be significantly reduced or waived. Approved sources play an important role in the biosecurity continuum, however, they must be regularly audited by desk-top studies and/or site visits to ensure ongoing compliance with Australia’s importing regulations.

Each year, more than four million plants are imported into Australia (pers. comm. A. Beutel, DAWE) which are classified as low, medium, or high risk depending on their significance as agricultural, horticultural, food or fibre crops, and their potential to carry harmful unwanted pests and diseases. Most of these imports are ornamental hosts imported as tissue culture plantlets in sterile vials. These plants present a low biosecurity risk and are inspected at the border, and if free of pests and disease symptoms are released. The next group of plants pose a medium biosecurity risk and require a minimum of three months in approved PEQ facilities (approved arrangements) undergoing pest treatment on arrival. Medium risk nursery stock consignments are inspected by a biosecurity officer for freedom from live insects, live snails, soil and disease symptoms of biosecurity concern. During growth in PEQ, all plants are inspected by a biosecurity officer for disease symptoms twice. Once during the active growing period and a final inspection no more than seven days prior to release. Samples from suspect plants are taken and analysed in a government biosecurity diagnostic laboratory by trained pathologists. The last group of plants are considered to pose the greatest risk because they can host significant pathogens of biosecurity concern that could have a negative impact on plant industries and/or the environment if they became established. Whilst these plants are imported in numbers less than 1000 plants per year, they undergo extensive periods of pathogen testing and, in some cases, pre-emptive pathogen treatment in Australian government PEQ facilities or accredited, industry specific PEQ facilities (e.g., sugarcane). High risk plants include potatoes, grapevines, berry crops, fruit trees, forest tree species, and ornamental hosts of high priority pests such as *Xylella*.

The Australian agricultural sector is heavily dependent on imported seeds and imports large quantities annually. Like plants, certain seed species require testing to ensure they are free of seed borne disease agents. If not tested pre export using an approved method, a sample must be tested on arrival at an approved laboratory of which there are two accredited providers in Australia; Plant Health Diagnostic Service (New South Wales Department of Primary Industries) and Crop Health Services (Agriculture Victoria). In some cases, seeds may also be grown in either a Government PEQ facility or, more commonly, at a DAWE approved arrangement site for screening and/or further testing. Specific import conditions for seed are listed in BICON.

As part of this special edition for MDPI on plant virology in Australia, this review paper provides a concise high-level historical overview on the evolution of diagnostics used for testing plant viruses in Australian government PEQ facilities.

## 2. Plant Virus Diagnostics Used in Government PEQ Facilities

More than 300 different plant genera are imported into Australia annually via PEQ. Australia conducts detailed risk analyses on imported nursery stock to identify what viruses pose a biosecurity risk. Viruses present in Australia and not under official control are removed from the list with the remaining viruses assessed for economic and environmental consequences. Viruses above Australia’s ALOP are deemed to pose a biosecurity risk and require testing in PEQ. The following plant viruses are of particular concern to Australia and host plants are tested in PEQ: Plum pox virus causes one of the most devastating diseases of stone fruit worldwide and spreads internationally through propagation material and locally by aphids [1]. Every *Prunus* consignment is screened with three independent tests (ELISA, bioassay and PCR) for plum pox virus before being released from quarantine. Other viruses of concern for high-risk plants include the nepoviruses cherry leaf roll, tomato ringspot, and raspberry ringspot and tomato black ring.

Viruses are screened using International Plant Protection Convention (IPPC) endorsed protocols. When an IPPC method is not available, preference is given to National Diagnostic Protocols (NDPs), which are developed in Australia by subject-matter experts and extensively reviewed and verified before receiving cross-jurisdictional endorsement. Equivalent internationally accepted methods such as the European and Mediterranean Plant Protection Organization (EPPO) or the National Plant Diagnostic Network (NPDN–United States Department of Agriculture) are also used. Where available, these protocols are used in PEQ to screen for high priority pests on any given consignment, and range from morphological/microscopic analysis through to serological and molecular diagnostic techniques. Diagnostic methods used in PEQ are referenced in BICON or documented in PEQ testing manuals. Given the frequency of updates, it is challenging to maintain a public list of assays however all diagnostic methods are freely available online or by request through the relevant regulatory agencies. Prior to discussing the different types of diagnostic approaches used in PEQ to test imported plants for viruses, the importance of appropriate sampling protocols must be emphasised. The accuracy of any diagnostic assay is largely dependent on prudent selection of samples with due consideration given to the plant tissue type selected and seasonality. Seasonality, however, is typically less of an issue with plants grown in temperature and light controlled PEQ glasshouse conditions compared to field grown plants. Due to potential for low titre or uneven distribution of viruses in different plant tissues, both young and mature leaves along with mid-rib samples are collected from imported plants usually in spring, as this period of vegetative growth is typically the most favourable for virus replication [2]. In most cases in PEQ, samples from individual plants are selected for diagnosis however it is possible to pool samples for pathogen testing from some plant species (maximum five plants/test). Details available in BICON. A related issue is the number of replicates required to give confidence in the testing. The number of replicate samples to be tested not only depends on the diagnostic assay employed but also on the target virus and the time of testing. In PEQ, hundreds of different viruses are tested annually and there are limitations on accessing positive and negative controls for all diagnostic tests. Whilst it is unquestionable positive and negative controls should be included in diagnostic tests, sometimes appropriate infected plant materials for pathogens of biosecurity concern are not available or cannot be created, particularly for in-vivo bioassay inoculations. In these cases, similar or synthetic positive controls are used. For negative controls, inoculation buffer alone or healthy sap in buffer is used for testing. Given these challenges and the need for importers to have safe access to new genetic stock in the shortest possible timeframe, PEQ diagnosticians have historically adopted a pragmatic approach regarding sampling protocols.

A broad range of traditional and modern diagnostic approaches are employed in PEQ facilities to index for viruses in imported plants. PEQ continues to adopt new technologies where applicable and retains older ones where needed. This review covers: (1) traditional, (2) current and (3) future diagnostic platforms for plant viruses.

## 3. Traditional Plant Virus Diagnostics

Each year, diagnosticians in government PEQ facilities conduct thousands of tests using a range of traditional diagnostic approaches including visual inspections for symptoms, bioassays, serology, and transmission electron microscopy to screen and test imported plants for the presence of viruses. Every plant released from a government PEQ facility is either tested for potential high risk pathogens, or is a clone of such a plant, propagated by government horticulturalists in the PEQ facility.

### 3.1. Visual Inspection

One of the historical reasons for using PEQ is to allow plants to undergo a period of active growth over extended periods of time (months to years), thereby allowing disease symptoms to develop if viruses are present. Diagnosticians regularly inspect every plant growing in PEQ for symptoms which can help narrow down the identity of a virus and allow removal of diseased plants. Visual inspection is relatively easy when symptoms are characteristic of a specific disease, however, many factors such as virus strain, host plant cultivar/variety, time of infection, and the environment can influence the symptoms exhibited. Plants can also exhibit virus-like symptoms as a response to unfavourable soil mineral/nutrient imbalances, infection by non-viral pathogens, damage caused by insect/mite/nematode pests, air pollution, or pesticides. Some viruses induce transient disease symptoms in the initial phase of infection but disappear with time. Other viruses produce no apparent symptoms at all. While symptoms provide useful information on virus diseases, using symptomatology alone is not sufficiently reliable, and it is necessary to perform additional confirmatory tests in PEQ to ensure presence/absence, and accurate diagnosis of virus infection.

### 3.2. Transmission Electron Microscopy

The Mickleham PEQ facility houses a modern transmission electron microscope (TEM) which provides a useful generic approach for detecting and narrowing down the identification of viruses based on the particle morphology observed [3,4]. Filamentous and rod-shaped viruses such as potyviruses, potexviruses, and tobamoviruses are readily differentiated in negatively stained leaf-dip preparations, whereas small isometric viruses such as nepoviruses are less readily observed. Likewise, low titre viruses in plant sap are not easily detectable unless the virus in the test material is concentrated before visualization. The TEM is useful for detection of new viruses and allows the recognition of mixed viral infections in a plant. There are limitations with using TEM as a diagnostic tool as identification is typically limited to family level unless immunosorbent electron microscopy (ISEM) is used. At Mickleham, ISEM is not routinely used, however it does offer the ability to identify viruses to species level.

### 3.3. Bioassays

Bioassays or biological indexing is one of the earliest active virus tests developed and has been used in PEQ for decades to detect plant viruses and other graft-transmissible pathogens (viroids, phloem-limited bacteria and phytoplasmas), particularly in fruit tree species, berry crops, grapevines and potatoes. The procedure is based on the ability of certain plants, called indicator plants, to produce symptoms when inoculated with viruses by grafting, budding or mechanical inoculation [5]. The development of symptoms on the indicator means the graft source material was infected with virus(es). Indicator plants are chosen for their ability to display relatively distinct disease symptoms when infected. Whilst bioassays are still used in plant quarantine because they can detect pathogens or strains that are not detected by more specific tests [6], there are many limitations with the method, not the least being that bioassays cannot usually identify viruses to species level and additional tests must be used to identify the virus. Additionally, the absence of symptoms on indicator plants is not proof of the absence of pathogens infecting the plant tested. In bioassays of grapevine, Constable et al. [7] found symptoms were not always observed, even when viruses were detected using molecular tests. Likewise, expression of symptoms on inoculated woody indicators is not always observed for fruit tree viruses, even when pathogens are detected by serological or molecular tests [8,9]. Bioassays are resource intensive requiring large amounts of expensive, temperature-controlled glasshouse space, experienced horticulturalists, and highly skilled practitioners (whose skills availability is declining globally) to interpret ‘disease’ symptoms. Bioassays are time consuming with some woody indicator testing for citrus viruses in PEQ taking more than two years before results are obtained.

In summary, most biological assays are not sensitive or specific enough to be used on their own. At best bioassays are a useful complementary tool but given their time constraints and subjectivity in interpreting symptoms, more timely, accurate, and reliable methods are needed in a phytosanitary setting.

### 3.4. Enzyme Linked Immuno-Sorbent Assay (ELISA)

ELISA serological testing was introduced into PEQ for screening plant viruses in the late 1980’s, and until recently was extensively used for detecting and identifying viruses in imported plants. The procedure is readily amenable to testing of pooled samples. ELISA assays are designed to detect a specific part of a viral protein and are generally more specific than TEM and bioassays, although some tests used in PEQ can detect a broad range of viruses at genus level making rapid group screening more cost-effective.

Whilst many of the traditional approaches to diagnosing plants for viruses are still employed in PEQ, these have been complemented or replaced with more modern molecular approaches.

## 4. Current Plant Virus Diagnostics

Polymerase chain reaction (PCR) testing was adopted for PEQ pathogen detection in the 1990’s and since then the range of both specific and generic assays has expanded significantly. Over 100 endpoint PCR and real-time (qPCR) assays are now routinely used to test high-risk plants commonly imported into Australia. These tests range from generic endpoint PCRs designed to detect many members of a virus family or genus to very specific qPCR assays which can detect specific strains of a virus. An ongoing challenge for PEQ molecular diagnosticians is the resource limitations to validate, standardise and automate such assays. With such a diverse range of assays across a similarly diverse number of target species, sampling and high-quality nucleic acid extraction continues to be a labour-intensive process due to the nuances associated with each host and pathogen regarding aspects such as extraction methods, targeted tissues and developmental stages.

Despite this, work is ongoing to introduce efficiencies, such as adopting high throughput automated DNA/RNA extraction platforms, 96 and 384 well PCR/qPCR systems, liquid handling robotics, etc. Even though significant investment in these automated systems has had success, PCR diagnostics in PEQ have now all but maximised their potential for efficiency gains with the technology currently available.

As identified pathogens continue to expand in number, range, hosts, virulence and diversity, and as new pathogens continue to emerge in parallel with the increasing trade in new plant genetics, a paradigm shift is required to meet the increasing demands of molecular testing.

## 5. Future Virus Diagnostic Platforms

As noted, PCR is still the backbone of PEQ virus diagnostics, and although modest efficiency dividends have been realised with the implementation of qPCR and automation, limitations exist with their implementation. There is a lack of internationally accepted and validated qPCRs available for biosecurity priority pests, and automation is limited in its scalability due to the large variety of commodities that undergo PEQ and the ever-evolving lists of pests requiring testing. In addition, PCR assays rely on prior knowledge of the plant pest’s genetic makeup to develop new tests, which is why molecular tests have traditionally been complemented using bioassays which provide a more generic approach for assessing presence of plant virus symptoms.

As much as PCR/qPCR, visual inspections, culturing, bioassays and ELISAs will continue to be important tools in the future, they are too labour-intensive to keep pace with the increasing demand on diagnostic capability/capacity in PEQ. With future forecasts predicting this trend will continue and with resources limited, staff numbers cannot be expected to increase in a manner commensurate with this growing trend.

Like all scientific fields, plant pathology and biosecurity are undergoing a quantum shift in technology as the golden age of ‘omics’ rolls on with no sign of abating. Since the discovery of the structure of DNA by Watson and Crick over 50 years ago, molecular diagnostics have developed and evolved at a staggering speed, becoming exponentially cheaper, faster, smaller, and more sensitive over time.

There is a powerful diagnostic tool on the horizon that will soon be ready for full implementation in PEQ. High Throughput Sequencing (HTS) is a mature technology in many research laboratories, but its power and scope make it challenging to deploy in a regulatory setting despite its obvious advantages. The main identified issue still requiring resolution is how to deal with putative, sequence-only potential pathogens. Previously a conservative approach was feasible due to the infrequency with which this occurred, but a universal roll-out will undoubtedly create issues from both a diagnostic and regulatory trade standpoint. Studies to date indicate most well-established commodities will encounter this issue rarely (Barrero, R. unpublished data), but for more exotic and less studied plant material it is likely such detections will occur more often. Without the time and resources available to generate the necessary biological data required to make an informed decision with sequence-only identifications, an objective, standardised and nationally accepted decision-making matrix is one potential solution to this issue. Another proposed solution is to limit bioinformatics output to only report regulated pathogens of concern, but this can still raise potential issues when known pathogen sequences are detected in unexpected hosts and/or countries of origins and may also miss emerging potential virus problems. Australia needs to have a robust, well defined, and defensible set of guidelines in place if challenged regarding HTS results, with these ideally developed in partnership with like-minded countries and trading partners. Any new diagnostic technology such as HTS undergoes a vigorous optimisation and validation process (often taking many years) including extensive side-by-side comparisons to assess the new technology against the existing diagnostic platforms. A DAWE expert panel comprised of plant pathologists, research scientists, policy regulators and IT professionals is currently developing a path forward to map a detailed plan for implementation of HTS in PEQ. The department has been conducting extensive comparisons of HTS against existing bioassays and PCR techniques used in PEQ since 2014. The expert panel relies on scientific data to clearly demonstrate how the benefits of HTS outweigh the potential risks noted above, as discussed by several authors in recent years from multiple jurisdictions [10,11,12,13,14], as well as findings from research already in progress in DAWE to directly compare the efficacy of HTS to existing PEQ testing platforms.

Looking over the horizon, the future of PEQ and biosecurity diagnostics more broadly will require innovative approaches that allow more to be done with less. Concepts already under development such as ‘Lab-in-a-tube’ and ‘Lab-on-a-chip’ will eventually mean in the future there will be less diagnostics undertaken in the laboratory and more carried out at the border. Lab-in-a-tube consists of the integration of various components of reactions into a consolidated microsystem that is portable, compact, and rapid [15]. This technology has advanced in the medical field to include rapid point-of-care assays for a range of human pathogens and diseases. Lab-on-a-chip consists of a single circuit that integrates one or more laboratory functions utilising microfluidics to automate analytical outcomes [16]. Lab-on-a-chip technology has likewise seen significant advances in diverse areas including chemical engineering [17], drug discovery and development [18,19], environmental monitoring, forensics, [20], and microorganism detection [21].

As these ‘point of care’ or in this case, ‘at the border’ technologies decrease in price and increase in speed, availability, acceptance, sensitivity, versatility and portability, the future world of plant biosecurity will follow the lead of human health and start adopting these and other as-yet unknown new diagnostics technologies. Whilst this will be good for biosecurity by increasing efficiency and thus reducing the turn-around time of results, care must be taken to ensure experienced, reliable, competent, and diverse diagnostics capability is retained within a PEQ setting. Even once such technologies are mature and widely accepted by regulators, biosecurity will always be part of a global biological system that is both inherently variable and constantly evolving. Experienced biosecurity research scientists and diagnosticians must be retained to monitor plant health, decipher ambiguous results, and develop, adapt, and optimise new assays as novel threats emerge and spread.

### Case Study

To highlight the benefits of adopting new virus diagnostic technologies, two case studies using Prunus and Fragariae are presented. In the late 1990s, *Prunus* plants imported into Australia spent a minimum of 27 months in PEQ (personal communication. M. Whattam, DAWE). This was due to the mandatory use of five woody bioassays which took considerable resources and time for symptom expression and interpretation. In 2005, PCR indexing was introduced into PEQ, and over the next few years additional new PCR tests were developed, validated and adopted, resulting in a significant reduction in the PEQ period to 16 months saving time and money for importers. Likewise, *Fragaria* imports into Australia took a minimum 24 months in PEQ and required four different University of California (UC) indicator plants and a range of herbaceous bioassays which required extensive time to express disease symptoms. With the introduction of new PCR protocols, the PEQ time period has been recently reduced to 12 months with a single bioassay performed using UC-5 indicator plants. The introduction of HTS as a routine diagnostic assay in PEQ is expected to further reduce the PEQ time. Future adoption of newer techniques and procedures including acceptance of offshore testing is likely to reduce time and costs still further in PEQ, thereby enabling Australia’s plant industries to be more internationally competitive, profitable and sustainable.

## 6. Conclusions

In conclusion, PEQ plays a significant role in the battle against entry of new plant viruses and strains into Australia. In the last decade, more than 150 exotic plant pathogens have been intercepted by DAWE including a number of significant virus pathogens (e.g., plum pox virus, grapevine corky bark virus complex, grapevine fan leaf nepovirus, citrus tristeza virus) that could have caused serious negative impacts on our plant health industries and environment. This highlights the importance of ensuring the best available science and diagnostic developments are constantly tested, evaluated, and implemented as they evolve.

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
