# Peer review of "Evolution of Plant Virus Diagnostics Used in Australian Post Entry Quarantine"

_plants, 2021, doi:10.3390/plants10071430_

Round 1
Reviewer 1 Report
Whattam et al. describe in their manuscript the diagnostic methods used in Australia during quarantine after entry of plant material. This is an important topic of interest not only in the country, but its application can be extended to other countries and regions that control the entry of plant material. Quarantine and diagnostic procedures are essential to eliminate or at least limit the entry of diseases and pests into the country.
However, this work, in my opinion, is very limited and should provide more information so that it can be of use and reference in other countries and regulatory agencies.
For example, no information is given on what control methods are now officially used in Australia for the diagnosis of the different viruses. For example, in Europe there is the EPPO which issues bulletins and diagnostic protocols. Is there an equivalent in Australia? Are the official diagnostic methods published? There is also not much detail given about the sampling of incoming plants. What proportion of plants are visually examined from a given lot? Of those inspected, what proportion of samples are diagnosed by ELISA, PCR, etc.? In addition to live plants, are seeds also tested? If so, what protocols are followed? Citing some examples would be very clarifying, especially if there are references.
The authors provide just a case study in Prunus, but I think that more examples should be given, given Australia's accumulated experience in controlling the entry of plant pathogens.
For example, in the case of grapevine, where the material replicates vegetatively and viruses are very abundant, what quarantine protocol is followed in the country? are bioassays performed? for how long?
Regarding horticultural plants where viruses are very important, what protocols are followed, since quarantine in this type of plants cannot be very long due to the conditions of the crop? Or maybe the importation of live plants of this kind is not allowed? If only seeds are allowed to enter, are the batches tested, are some seeds germinated for analysis?
In ornamental plants, which are not intended for cultivation and multiplication but for distribution to markets and florists, quarantine cannot be very long because plants deteriorate very quickly. In this case, what protocol is followed? visual inspections? Is there any sampling for diagnosis? This is just to mention a few examples, but many more interesting cases are surely available.
In the case that new diagnostic techniques such as HTS are to be implemented in post entry control, how are these types of decisions made? Is there an expert panel? Are comparison HTS/Bioassays/PCR studies in progress?
In short, I find the manuscript interesting but it should be more ambitious to be of more general interest.
Author Response
Reviewer’s comments and responses.
Thank you for taking your valuable time to review and provide comments on the manuscript. We’ve updated the manuscript to reflect some but not all your comments. The intent of the review is to provide a concise high-level overview of the evolution of virus diagnostics used in Australian post entry quarantine without providing too many details otherwise the paper will become a very large document. However, the suggestions are worthy of a separate paper to provide more details on virus diagnostics and target pests. The authors will consider preparing a separate paper in due course. If this additional detail is required for publication of the manuscript a significant time extension will be required from the journal editor. Please see responses in purple.
Reviewer 1.
This is an important topic of interest not only in the country, but its application can be extended to other countries and regions that control the entry of plant material. Quarantine and diagnostic procedures are essential to eliminate or at least limit the entry of diseases and pests into the country. However, this work, in my opinion, is very limited and should provide more information so that it can be of use and reference in other countries and regulatory agencies. For example, no information is given on what control methods are now officially used in Australia for the diagnosis of the different viruses. For example, in Europe there is the EPPO which issues bulletins and diagnostic protocols. Is there an equivalent in Australia? High priority pests are screened using International Plant Protection Convention (IPPC) endorsed protocols. When an IPPC method is not available for a targeted pest, preference is given to National Diagnostic Protocols (NDPs), which are developed in Australia by subject-matter experts and extensively reviewed and verified before receiving cross-jurisdictional endorsement. Equivalent internationally accepted methods such as the European and Mediterranean Plant Protection Organization (EPPO) or the National Plant Diagnostic Network (NPDN – United States Department of Agriculture) are also used. Where available, these protocols are used in PEQ to screen for high priority pests on any given consignment and range from morphological/microscopic analysis through to serological and molecular diagnostic techniques. All methods are freely available online through the relevant regulatory agencies.
Are the official diagnostic methods published? Diagnostic methods used in PEQ are referenced in BICON or documented in PEQ testing manuals. Given the frequency of updates, it is challenging to maintain a public list of assays however they are freely available on request.
There is also not much detail given about the sampling of incoming plants. What proportion of plants are visually examined from a given lot? On arrival, all nursery stock consignments are inspected by a biosecurity officer for freedom from live insects, live snails, soil and disease symptoms of biosecurity concern. During growth in PEQ, all plants are inspected by a biosecurity officer for disease symptoms twice. Once during the active growing period and a final inspection no more than seven days prior to release. Samples from any suspect plants are taken and analysed in a biosecurity diagnostic laboratory by trained pathologists.
Of those inspected, what proportion of samples are diagnosed by ELISA, PCR, etc.? It depends on the symptoms and host, diagnosticians use ELISA, PCR or other techniques such as TEM.
In addition to live plants, are seeds also tested? Most seed are visually assessed at the border and released however some species pose a greater biosecurity risk and require active testing. Import conditions for seed are listed in BICON.
If so, what protocols are followed? Difficult to answer as depends on many factors and what we believe are the best diagnostic platform to use. As noted above the manuscript is a high-level overview and not a detailed analysis of every diagnostic test used on every host otherwise it would extend the paper to many pages.
Citing some examples would be very clarifying, especially if there are references. The authors provide just a case study in Prunus, but I think that more examples should be given, given Australia's accumulated experience in controlling the entry of plant pathogens. For example, in the case of grapevine, where the material replicates vegetatively and viruses are very abundant, what quarantine protocol is followed in the country? are bioassays performed? for how long?Agree - an additional case study is cited using strawberry genetics noting only high-level details are provided rather than details of every diagnostic test used (there are many hundred).
Regarding horticultural plants where viruses are very important, what protocols are followed, since quarantine in this type of plants cannot be very long due to the conditions of the crop? As indicated above, the review is not intended to provide detailed protocols of every test or virus as that would increase the size of the manuscript.
Or maybe the importation of live plants of this kind is not allowed? If only seeds are allowed to enter, are the batches tested, are some seeds germinated for analysis? In ornamental plants, which are not intended for cultivation and multiplication but for distribution to markets and florists, quarantine cannot be very long because plants deteriorate very quickly. In this case, what protocol is followed? visual inspections? Yes, and in some cases propagatable cut flowers such as roses are devitalised using herbicides however this is not the intent of the paper. Is there any sampling for diagnosis? Yes, in most cases single plants are selected for diagnosis however it is possible to pool some plants (maximum five plants/test) for some pathogens. This is just to mention a few examples, but many more interesting cases are surely available. In the case that new diagnostic techniques such as HTS are to be implemented in post entry control, how are these types of decisions made? Any new diagnostic technologies such as HTS undergo a vigorous optimisation and validation process (often taking many years) including extensive side-by-side comparisons to assess the new technology against the existing diagnostic platforms. Is there an expert panel? Yes, both external and internal subject matter experts are engaged in developing and optimising new technologies. Are comparison HTS/Bioassays/PCR studies in progress? Yes, the department has been investigating the use of HTS since 2014 and conducting extensive comparisons of HTS against existing techniques and are progressing development of a regulatory policy to use HTS for routine use in PEQ. In short, I find the manuscript interesting but it should be more ambitious to be of more general interest. Please note this review is part of a special edition on Australian plant virology and it’s not possible to address all the above comments in a high-level overview otherwise it will become a very large manuscript. However, the suggestions are worthy of a separate paper to provide more details on virus diagnostics and target pests. The authors will consider preparing a separate paper in due course.
Reviewer 2 Report
The manuscript consist of an interesting review on the strategies for plant virus detection applied by the Post Entry Quarantine (PEQ) service in Australia.
The paper deals with a topic of interest and supplies an exhaustive description about the historical application of traditional and novel diagnostic procedures for the evaluation of the sanitary status of imported plants related to virus presence.
Moreover, the Authors efficiently highlight and compare the current diagnostic tools with the novel available diagnostic approaches based on the high throughput sequencing technology.
The review is concise and well written. I think it is a worthy paper with interesting information to report.
However, some improvements should be made to the manuscript before publication:
- The description of methods reported in the Section 3 and 5 should be supported by the citation of proper literature.
- Authors should describe more than one case study to better highlight the benefits achieved by PEQ through the adoption of innovative methods for diagnosing the presence of plant viruses in plants imported from abroad.
Author Response
Reviewer’s comments and responses.
Thank you for taking your valuable time to review and provide comments on the manuscript. We’ve updated the manuscript to reflect some but not all your comments. The intent of the review is to provide a concise and high-level overview of the virus diagnostics used in Australian post entry quarantine without providing too many details otherwise the paper will become a very large document. However, the suggestions are worthy of a separate paper to provide more details on virus diagnostics and target pests. The authors will consider preparing a separate paper in due course. If this additional detail is required for publication of the manuscript a significant time extension will be required from the journal editor. Please see responses in purple below. Thank you.
The paper deals with a topic of interest and supplies an exhaustive description about the historical application of traditional and novel diagnostic procedures for the evaluation of the sanitary status of imported plants related to virus presence. Moreover, the Authors efficiently highlight and compare the current diagnostic tools with the novel available diagnostic approaches based on the high throughput sequencing technology. The review is concise and well written. I think it is a worthy paper with interesting information to report. However, some improvements should be made to the manuscript before publication: The description of methods reported in the Section 3 and 5 should be supported by the citation of proper literature. Thank you, the manuscript has been updated with additional citations.
Authors should describe more than one case study to better highlight the benefits achieved by PEQ through the adoption of innovative methods for diagnosing the presence of plant viruses in plants imported from abroad. Agree an additional case study using strawberry plants has been included. Note whilst additional case studies could be added the review attempts to provide a high-level overview of Post Entry Quarantine and be concise.
Reviewer 3 Report
In the manuscript “Evolution of plant virus diagnostics used in Australian Post En-2 try Quarantine” the authors present a review on the the diverse diagnostic techniques used by the Australian government Post En-9 try Quarantine (PEQ) facilities for testing the presence of viruses in the imported plants for viruses.
1-The paper is nicely written and presented and no doubt it provides an interesting information about the Australian plant virus detection strategy. However, as a reader, I would like to have some more specific details on which viruses are the most serious threats and are the main targets of the diagnostic techniques.
Could be possible that the authors indicate in the introduction section some of the main viruses that the Australian government is most concerned which could be introduced in the country and are actively searched when testing imported plants?
2-There are two main groups of diagnostic techniques. Those that try to detect the presence of any virus in the sample such as, visual inspection, transmission electron microscopy, bioassays or high throughput sequencing and the ones that are designed to detect specific plant viruses such as ELISA or PCR. I mean that you have to decide first which viruses you want to check if they were in the sample and then would chose specific antibody or pairs of primers.
In ELISA test which are the main viruses searched?
In the PCR section I think the RT-qPCR technique should be also indicated since most plant viruses have RNA genomes. Finally, could the authors report some examples of specific virus strain or genera considered as serious agronomical threats for the country
Author Response
Thank you for taking your valuable time to review and provide comments on the manuscript. We’ve updated the manuscript to reflect some but not all your comments. The intent of the review is to provide a concise high-level overview of the virus diagnostics used in Australian post entry quarantine without providing too many details otherwise the paper will become a very large document. However, the suggestions are worthy of a separate paper to provide more details on virus diagnostics and target pests. The authors will consider preparing a separate paper in due course. If this additional detail is required for publication of the manuscript a significant time extension will be required from the journal editor. Please see responses in purple below. Thank you.
1. The paper is nicely written and presented and no doubt it provides an interesting information about the Australian plant virus detection strategy. However, as a reader, I would like to have some more specific details on which viruses are the most serious threats and are the main targets of the diagnostic techniques. Could be possible that the authors indicate in the introduction section some of the main viruses that the Australian government is most concerned which could be introduced in the country and are actively searched when testing imported plants? Thank you for the suggestion, the manuscript has been updated to include key virus risks. Australia conducts detailed risk analyses on imported nursery stock to identify what viruses may be present in the host. Viruses that are already present in the country are removed and as assessment of the consequence of the remaining viruses are determined. Viruses assessed as being of biosecurity concern are then actively screened in PEQ. More than 300 different plant genera are imported into Australia annually via PEQ. The following viruses are of particular concern to Australia and host plants are tested in PEQ: Plum pox virus, one of the most devastating diseases of stone fruit worldwide that spreads through propagation material and aphids [1]. Every Prunus consignment is screened with three independent tests for Plum Pox before being released from quarantine. Other viruses of concern for high-risk plant commodities include the nepoviruses cherry leaf roll, tomato ringspot, and raspberry ringspot and tomato black ring.
2. There are two main groups of diagnostic techniques. Those that try to detect the presence of any virus in the sample such as, visual inspection, transmission electron microscopy, bioassays or high throughput sequencing and the ones that are designed to detect specific plant viruses such as ELISA or PCR. I mean that you have to decide first which viruses you want to check if they were in the sample and then would choose specific antibody or pairs of primers. In ELISA test which are the main viruses searched? This very much depends on the host and list of viruses of biosecurity concern. Given the manuscript is providing a high level and concise overview this detail would greatly expand the paper. As indicated above, the authors will consider preparing a more detailed manuscript on viruses of biosecurity concern and diagnostic techniques employed in a future manuscript. In the PCR section I think the RT-qPCR technique should be also indicated since most plant viruses have RNA genomes. Finally, could the authors report some examples of specific virus strain or genera considered as serious agronomical threats for the country. As indicated above specific viruses of biosecurity concern have been added to the revised manuscript. Thank you.
Round 2
Reviewer 1 Report
After the revision made by the authors I believe the manuscript can now be published.